# Five-year-olds' facial mimicry following social ostracism is modulated by attachment security

Stefania V. Vacaru[1]*, Johanna E. van Schaik[2], Erik de Water[3], Sabine Hunnius[1]

1 Donders Institute for Brain, Cognition, and Behaviour, Radboud University, Nijmegen, The Netherlands,
2 Vrije Universiteit, Amsterdam, The Netherlands, 3 University of Minnesota, Minneapolis, MN, United States of America

* v.vacaru@donders.ru.nl

**Data Availability Statement:** Data are available from the Donders Institute repository. All data, except identifiable information has been shared. The link to the repository is https://data.donders.ru.nl/collections/di/dcc/DSC_2018.00151_758.

## Abstract

Social ostracism triggers an increase in affiliative behaviours. One such behaviour is the rapid copying of others' facial expressions, called facial mimicry. Insofar, it remains unknown how individual differences in intrinsic affiliation motivation regulate responses to social ostracism during early development. We examined children's facial mimicry following ostracism as modulated by individual differences in the affiliation motivation, expressed in their attachment tendencies. Resistant and avoidant tendencies are characterized by high and low affiliation motivation, and were hypothesized to lead to facial mimicry enhancement or suppression towards an ostracizing partner, respectively. Following an ostracism manipulation in which children played a virtual game (Cyberball) with an includer and an excluder peer, mimicry of the two peers' happy and sad facial expressions was recorded with electromyography (EMG). Attachment was assessed via parent-report questionnaire. We found that 5-year-olds smiled to sad facial expressions of the excluder peer, while they showed no facial reactions for the includer peer. Neither resistant nor avoidant tendencies predicted facial mimicry to the excluder peer. Yet, securely attached children smiled towards the excluder peer, when sad facial expressions were displayed. In conclusion, these findings suggest a modulation of facial reactions following ostracism by early attachment.

## Introduction

Human behaviour is shaped by the need to belong and hence the intrinsic drive to affiliate with others [1, 2]. To achieve affiliation, individuals use subtle strategies from early on in development [3–6]. One such behaviour is the rapid copying of others' facial expressions, commonly referred to as facial mimicry [7, 8]. Facial mimicry is thought to play an important role in interpersonal relationships, as it has been shown to increase liking and foster affiliation amongst social partners [9, 10]. Furthermore, it has been shown that facial mimicry is modulated by social context [11]. For example, individuals mimic more in-group compared to out-group members [5, 12] and people towards whom they hold positive attitudes [13].

**Funding:** This research was supported by an Aspasia Prize of the Netherlands Organisation for Scientific Research (NWO) awarded to SH. The funder had no role in study design, data collection and analysis, decision to publish, or preparation of the manuscript.

**Competing interests:** : The authors have declared that no competing interests exist.

Furthermore, recent evidence suggests that one's intrinsic motivation for affiliation modulates facial mimicry [14]. Facial mimicry of emotional expressions is not merely the result of activation matching, but is also influenced by one's affect. Contrary to postures, facial emotions are inherently meaningful and their mimicry engage affective and motivational processes [15–17]. Accordingly, inner states and intrinsic characteristics (e.g. attachment orientation [14]; callous-unemotional traits [18], in concert with affiliation motives [19]; power motivation [20] modulate the extent to which individuals manifest facial mimicry in social contexts [21, 22]).

## Ostracism

Particularly when one's belonging is under threat, individuals strive to restore positive feelings by recruiting affiliative behaviours [23, 24]. For instance, children who had been primed with ostracism showed greater affiliation motivation by drawing characters closer to each other [25] and showing more prosocial behaviours [26] as well as higher imitation fidelity in a subsequent behavioural task as means to convey liking and similarity to a social partner [27–29]. In the same vein, following social exclusion in an experimentally manipulated ball-tossing computer game (i.e. Cyberball; [30–32]) children showed more affiliative high-fidelity imitation behaviours [33, 34].

## Individual differences in the motivation for affiliation

When investigating the effects of ostracism on affiliative behaviours, it is crucial to bear in mind that although a universal human drive–the need to belong–is triggered [1, 25], humans may differ considerably in their sensitivity to social exclusion and their intrinsic motivation to affiliate with others. Early attachment for instance has a pervasive effect on one's social motivation. From early on, children form affective ties with their caregivers [35, 36], and these attachment relationships are thought to constitute the basis for their socio-emotional development [37]. When children's needs are adequately met by their primary caregivers, they develop secure trusting relationships, influencing the quality of other relationships in childhood and later in life [38–40]. Based on the quality of early attachment relationships, children form a secure or an insecure attachment style. Particularly, childhood insecure attachment can be distinguished into resistant/ambivalent (also called preoccupied in adulthood) and avoidant (also called dismissing in adulthood) [41]. While *resistant* attachment is characterized by an anxious style and high motivation for affiliation as a means of maximizing the chances for proximity and feelings of acceptance, *avoidant* attachment is characterized by a dismissing style and diminished motivation for affiliation [42, 43].

Accordingly, some evidence indeed shows that early *resistant* attachment tendencies yield enhanced facial mimicry, suggesting higher intrinsic affiliation motivation, irrespective of the social context [14]. Another study found that adults with an *avoidant* attachment style tend to suppress facial mimicry responses to negative expressions (i.e. anger facial expressions) and instead react with an opposite expression (i.e. happy facial expression), which has been interpreted as an attempt to suppress negative emotion processing [44]. Together, these studies suggest that individual differences in insecure attachment tendencies influence the situational motivation for affiliation and, hence, facial mimicry.

While there is limited work examining behavioural reactions to social ostracism and the effect of attachment tendencies on these affiliative behaviours, neurocognitive studies support the hypothesis that individual differences in one's motivation for affiliation affect one's reactions to ostracism. In a functional Magnetic Resonance Imaging (fMRI) Cyberball study investigating the relation between attachment styles and neural responses to social ostracism [45], individuals scoring high on *resistant* attachment showed increased neural activation in

response to social rejection in the dorsal anterior cingulate cortex and anterior insula, indicating greater negative responses to rejection and potentially greater distress. In contrast, individuals scoring high on *avoidant* attachment showed considerably dampened neural activation in those areas, suggesting that avoidant individuals may be emotionally distancing themselves from distressing interactions with others, as a way to keep a safe distance from others [45]. Likewise, an fMRI study investigating neural responses to social exclusion in children with early life separation experiences found reduced neural activity in areas implicated in emotion regulation, and children reported more feelings of exclusion and frustration during the Cyberball manipulation compared to controls [46]. Furthermore, in adults, it has been shown that secure attachment buffers the effects of social exclusion during a Cyberball game, as revealed by diminished neural activation in brain regions implicated in emotion regulation [47]. Taken together, these findings highlight differences in the sensitivity to social rejection between individuals with different attachment styles, which possibly may also guide individuals' affiliative behaviours during interactions characterized by social exclusion.

## The current study

In light of earlier findings suggesting that early life experiences with caregivers greatly affect the neural and behavioural responses to social ostracism, we examined young children's affiliative behavioural responses, as measured by facial mimicry, following ostracism and their modulation by early attachment tendencies. Accordingly, the aim of this study was twofold.

First, we investigated children's affiliative behaviours by means of their facial mimicry of peers with whom they had engaged in a Cyberball game. We used two exemplars of positive and negative emotions, similar to the previous studies [14, 44]. Particularly, we chose sad instead of angry emotional expressions, given that angry expressions may elicit a fear reaction rather than a mimicry response [48]. More specifically, using electromyography (EMG), we assessed whether children would display increased zygomaticus major (ZM; smiling) and decreased corrugator supercilii (CS; frowning) muscle activation in response to happy facial expressions, and conversely, whether they would display increased CS and decreased ZM muscle activation in response to sad facial expressions. We examined whether children's facial responses differed for a peer who had played the game in an unkind, excluding way and a peer who played in a kind, including way. In line with previous evidence that showed an increase in affiliative imitation behaviours following ostracism [28, 33], we expected that children would display stronger facial mimicry to the excluder's compared to the includer's facial expressions. In other words, being ostracized will trigger their motivation to restore positive feelings and hence to affiliate with the excluder peer by means of facial mimicry, while this will not be the case for the includer peer, with whom the interaction was already of an affiliative inclusive nature. Furthermore, we expected a stronger modulation of sad compared to happy expressions. This hypothesis is in line with previous findings showing no differences between inclusion and exclusion for happy facial mimicry, potentially due to the high automaticity of facial mimicry of happy expressions irrespective of the social context [49].

Second, we examined the modulation of children's facial responses to the excluder peer by attachment tendencies. According to the attachment theory, attachment behaviours become manifest under conditions of distress [50], which in this experiment is induced through the experience of ostracism. Attachment threat paradigms have been commonly used in attachment research to activate the attachment behavioural system and study the effects of different attachment tendencies on social dynamics, including mimicry [43, 51, 52]. Similarly, we expected that the effect of ostracism would be particularly salient in children characterized by insecure attachment, and resistant and avoidant tendencies would lead to different behavioural

strategies. Moreover, as the attachment system is activated following a distressing experience, we expected that insecurely attached children would exhibit distinct dis/affiliative behaviours, aimed at restoring positive feelings in relation to the source of distress. Consequently, in line with previous evidence [14, 36], we hypothesized that children characterized by resistant attachment would show increased facial mimicry to the excluder, as opposed to children characterized by avoidant attachment who were expected to show decreased facial mimicry, as means to increase or decrease chances for affiliation, respectively [23]. Hence, we assessed resistant and avoidant attachment tendencies as they lead to distinct strategies to attain affiliation.

## Method

### Participants

Participants were recruited from a database of volunteer families of the Baby and Child Research Center, Radboud University, Nijmegen, The Netherlands. Forty-two children participated in the study, but eight of them were excluded due to children not accepting the electrodes on their face ($n = 1$) and not having the minimum numbers of trials (at least two per peer and emotion) after EMG artefact rejection ($n = 7$). Thus, a total of 34 children (21 girls; $M_{age} = 4.72$ years, $SD_{age} = .34$ years, range = 3.81–5.27 years) were included in the final analyses. The sample size was determined based on prior work using a similar approach in which a medium-sized effect ($R^2 = .39$) of attachment on facial mimicry was found [14, 27]. Accordingly, a power analysis was performed using a linear multiple regression from the t-test family, two-tailed, three predictors (i.e. resistant, avoidant attachment and their interaction) assuming a medium effect size of .39, with .95 power and $\alpha$ of 0.05. This calculation showed a required sample of 36 participants. (G*Power software; [52]). Written informed consent was given by parents prior to their children participation in the study. Ethical approval for the study was obtained from the Ethics Committee of the Faculty of Social Sciences, Radboud University (ECSW-2017-1301-470). The study was conducted according to the ethical standards of the Declaration of Helsinki.

### Procedure

The session started with a 10-minute warm-up phase during which the child became acquainted with the testing room and the experimenters. Meanwhile, parents were informed about the study and were asked for their written informed consent. When the child seemed at ease, the experimenter asked whether they liked to play with stickers. The experimenter let the child select a cartoon character (e.g. Mickey Mouse) printed on an A4 sheet, which had small rectangles drawn on their faces, corresponding to the EMG electrode arrangement. The child was first encouraged to place stickers in the rectangles on the cartoon character's face and was then asked whether they would like to have similar stickers on their own face. The experimenter then asked the child to take a seat in front of a screen (60 cm distance) and explained that they would also receive stickers (i.e. the electrodes) on their face in the same spots as the rectangles on the cartoon sheet. Once the electrodes were attached to the skin, the experimenter checked the impedances and the EMG signal quality. When the signal quality was not within the 100 uV range or impedances were too high (> 10 kOhm), the skin was cleaned more and conductive gel was added to the electrodes, if necessary. Once the electrodes were attached well, the experiment started.

Children were instructed that they would use the buttons to toss the ball to the other players. First, they played the familiarization phase with the cartoon-like players. Then, the ostracism manipulation started, in which children were told that they would play a game with two

other children who were in different rooms. The pictures of the two players displaying a neutral facial expression were shown on the left and the right side of the screen. After the ostracism manipulation, the mimicry task started in which participants were shown pictures of the two players portraying happy and sad facial expressions. Finally, to restore positive feelings, children played another round of the Cyberball game with the two players now portraying happy facial expressions and both playing fairly.

Children's behaviour during the task was monitored via a camera. The experiment was interrupted if the child did not want to continue. After the EMG recording, the electrodes were removed and the child's face was cleaned with a wet wipe to remove any leftover gel. At the end of the experiment, children were compensated with 10 euros or a book.

### Attachment security

Attachment security was examined with the *Attachment Insecurity Screening Inventory 2–5* (AISI) [53, 54]. The AISI is a parent-report measure used to assess attachment of children between 2 and 5 years of age. Its reliability and validity have been examined across typical and clinical populations, indicating sound psychometric properties in terms of convergent, concurrent, and predictive validity as well as discriminating power between secure and insecure preschoolers [43]. The questionnaire contains 20 items on a 6-point Likert scale that belong to 3 subscales: avoidant, ambivalent/resistant and disorganized attachment. For the purpose of this study, only avoidant and resistant attachment tendencies were used in the analyses, as continuous scores. According to discriminating power analyses, a cut-off score of 46 from the total scale distinguishes between securely ($< 46$) and insecurely attached ($> 46$) children [53, 54]. Accordingly, 59% of children qualified as securely attached and 41% as insecurely attached, in line with previous findings on the prevalence rate of attachment security distribution in the general population [14, 54, 55]. Internal consistency analysis yielded a Cronbach's $\alpha$ of .735 for the avoidant subscale, .747 for the ambivalent/resistant subscale and .781 for the total scale, similar to previous results [34, 43].

### Cyberball paradigm

The Cyberball paradigm [30, 32] was used to manipulate social dynamics of inclusion and exclusion. This experimental paradigm consists of a computer game in which the participant and two virtual players toss a ball to each other (Fig 1). Children were told that they would play a game with two other children through the computer, while they were all in separate rooms (the participant in the blue room and the other players in the green and the orange room). The participants used a two-button response pad on which they placed both hands and pressed the right button with the right hand and the left button with the left hand to toss the ball either to the player on the right or on the left side of the screen. Social dynamics were manipulated by programming the computer players to toss the ball to the participant either just once (excluder) or in a fair manner (includer). The number of ball tosses of the includer player was dependent on the tosses of the participant and to whom they tossed the ball first, as the includer had to toss the ball an equal number of times to each player. Overall, children tossed the ball 7.39 times ($SD = 4.36$) to the includer and 8.58 times ($SD = 2.17$) to the excluder, $t(32) = 1.24$, $p = .222$.

Participants were first familiarized with the game by playing with two cartoon characters who tossed the ball equally often to each other and to the child. Afterwards, the ostracism manipulation started in which the pictures of two players were displayed each on one side of the screen in a coloured square (orange or green) representing the room where they were supposedly playing from. The players were male child models for boy participants and female

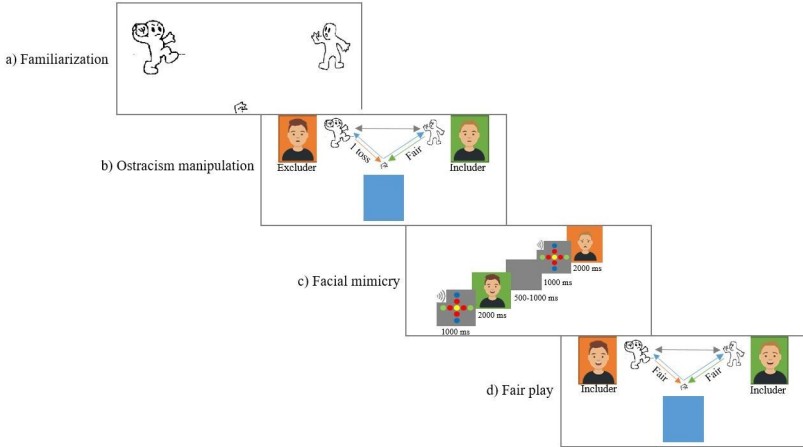

**Fig 1. The figure illustrates the four phases of the study.** In the first phase (a), children were familiarized with the game. In the second phase (b), children were presented with the ostracism manipulation, in which the virtual players were pre-programmed, such that the excluder only tossed the ball once to the participant, while the includer played fairly. In the third phase (c), FM to the excluder and the includer was assessed, by displaying pictures of happy and sad facial expression of the two players. The last phase (d) of the experiment consisted of another round of the game, in which both players played fairly, and served the purpose of restoring positive feelings. The avatars in this figure are not the original images used in the study, which could not be published due to the copyright license of the stimuli from the Radboud Faces Database. The avatars were created by the authors of this study and used here for illustrative purposes only (https://getavataaars.com.

child models for girl participants. The side of the screen (right, left) as well as the room colours (green, orange) in which the models were presented were counterbalanced across participants. A blue square at the bottom of the screen represented the room where the participants were playing. At the end, children played another round in which both players tossed the ball equally frequently to the participant to restore positive feelings. Altogether, the game lasted approximately 5 minutes.

## Stimulus material and facial mimicry paradigm

For the Cyberball and facial mimicry paradigms, images (240 x 180 pixels) of White female and male children (four in total) were selected from the Radboud Faces Database [56]. During the Cyberball game, the models had a neutral facial expression.

In the facial mimicry task, happy and sad facial expressions of both models were repeated eight times in a pseudo-randomized order, with the constraints that the same peer displaying the same emotion was never repeated right after each other. Hence, a total of 32 pictures were presented (i.e. 2 models x 2 facial expressions x 8 repetitions). Each trial lasted approximately 4000 ms and unfolded as follows: 1000 ms fixation cross, 2000 ms picture presentation and a jittered inter-trial interval of 500 to 1000 ms. With the onset of the fixation cross a short beep was played as an attention-getter. The tasks were displayed on a 17" monitor (1280 x 1024 pixels) and watched from a distance of 60 cm.

## EMG recordings

EMG responses were measured with Brain Vision Recorder [57]. Pediatric disposable 4-mm Ambu-Neuroline 700 Ag/AgCl surface electrodes were used to record muscle activation from the ZM and CS muscles with a bipolar configuration and 10 mm inter-electrode distance between their centres [15, 58]. The ground electrode was attached on the forehead below the hairline, and the reference electrode was attached on the mastoid bone behind the ear. The

guidelines for optimal placement of bipolar surface EMG electrodes in the face were followed [59], consistent with previous studies with young children [14, 48, 60]. A sampling rate of 2500 Hz was used, and a low cut-off filter of 10 Hz and a high cut-off of 1000 Hz were set for the data acquisition. To ensure good quality data acquisition, the standard procedures for EMG muscle site preparation and placement were followed [61]. The skin over the muscle group was cleaned and using Nuprep Skin Prep Gel and baby cleanser wipes. Moreover, conductive OneStep clear gel was added to the already pre-gelled electrodes to improve their impedances.

## EMG data processing

EMG data were pre-processed and normalized with Brain Vision Analyzer 2.1 [57], following the recommendations of De Luca [62]. First, the trials were filtered using a notch filter with band rejection of 50 Hz, 0.2 bandwidth, order 4, as implemented in BVA. Next, an infinite impulse response zero phase shift Butterworth filter with a low cut-off frequency of 20 Hz and high cut-off frequency of 500 Hz, and a 12 dB/octave slope was applied [62]. After the data pre-processing, artefact rejection based on visual investigation of the EMG signal was conducted. The signals of both muscles were screened between 500 ms prior and the 2000 ms after stimulus onset, hence for segments with a total duration of 2500 ms. The segments were inspected for extreme amplitude values outside a 100 mV range. If any peaks during a segment indicated such extreme values, the trial was rejected. The mean number of trials after pre-processing was 26.5 ($SD$ = 6.18, $Min$ = 12, $Max$ = 32); with on average 6.76 ($SD$ = 1.48; $Min$ = 3, $Max$ = 8) trials for the happy includer, 6.65 ($SD$ = 1.97, $Min$ = 2, $Max$ = 8) for the happy excluder, 6.32 ($SD$ = 2.10; $Min$ = 2, $Max$ = 8) for the sad includer, and 6.76 ($SD$ = 1.41; $Min$ = 4, $Max$ = 8) trials for the sad excluder peers. The difference in the number of trials for the peer-emotion combinations was not significant, $F(1, 33)$ = 2.11, $p$ = .156. Lastly, the signals were rectified.

EMG data was standardized within participants and within muscles to allow comparisons across different muscles [58]. To this end, data was divided in bins of 100 ms and each bin was standardized by subtracting from each value the mean activation in microvolts of all the bins and dividing it by the standard deviation of all bins. Next, we calculated the mean of the baseline bins and performed the baseline correction by subtracting the baseline mean activation from each 100 ms bin of the time window of the stimulus presentation. Thereafter, we calculated the mean activation of the whole 2000 ms time-window for subsequent analyses. The time course of standardized muscle activation in the CS and ZM muscles over the trial for the two different emotions and peers is displayed in Fig 2.

## Statistical analyses

To address our first aim, namely to investigate children's affiliative behaviours by means of their facial mimicry of peers with whom they had engaged in a Cyberball game, we tested whether children mimicked happy and sad facial expressions of an includer and an excluder peer. This was done with a within-factors repeated measures ANOVA with emotion (happy, sad), muscle (CS, ZM) and peer (includer, excluder) as independent variables and mean standardized muscle activations as the dependent variables. This analysis provides information as to whether muscle activation differences exist as a function of the observed emotional expression, when it is displayed by one or the other peer.

Second, to test whether avoidant or resistant attachment tendencies modulate facial responses to the excluder peer, we ran two linear regression models with happy and sad facial mimicry as the dependent variables and avoidant and resistant attachment continuous scores as predictors. To this end, composite scores for happy and sad mimicry were created to index

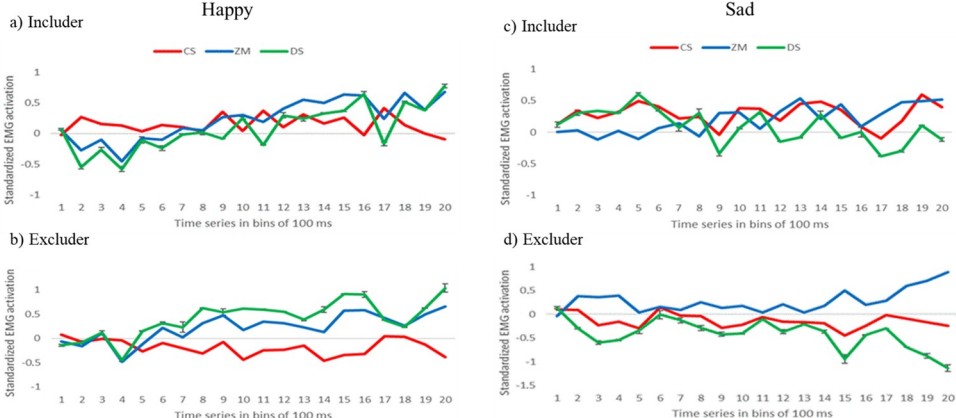

**Fig 2.** Time course of standardized EMG activation of the CS (red) and ZM (blue) muscles in response to happy includer (a), happy excluder (b), sad includer (c) and sad excluder (d) in bins of 100 ms from stimulus onset (0 ms) to stimulus offset (2000 ms). The green line represents the difference score between the two muscles (ZM and CS), in which each bin represents the mean (with the standard error) amplitude over 100 ms time.

the response to an observed expression, rather than activation of each muscle as a function of the observed emotion. Using one score as an index of facial mimicry reduces the complexity of the analysis and eases interpretation of effects.

## Results

### Facial mimicry following ostracism

The summary statistics of the muscles for each emotion and peer, and their intercorrelations are displayed in S1 Table. Fig 3 illustrates the violin plots for the happy and sad facial expressions for the includer and the excluder peer.

The repeated measures ANOVA revealed a non-significant three-way interaction between emotion (happy, sad), muscle (ZM, CS), and peer (includer, excluder) $F(1, 33) = 1.01$, $p = .323$. Further, a two-way interaction between muscle and peer emerged, $F(1, 33) = 10.16$, $p = .003$,

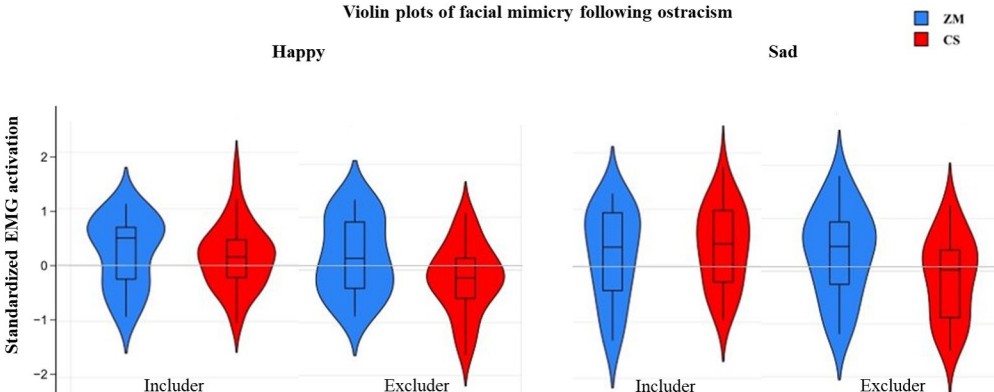

**Fig 3. Violin plots illustrating the muscle activation in the ZM (blue) and the CS (red) in response to happy and sad expressions for the includer and the excluder peer.** The distribution of the data is represented by the violin shape, with larger width indicating higher value frequency. The mean of each muscle activation for each peer is represented by a plus sign, whereas the horizontal bars represent the minimum, the median and the maximum values. The whiskers represent the first and the fifth quantile. ns = non-significant; $^*$ $p < .05$.

$\eta p^2$ = .23, suggesting differences in the muscle reactions for the includer and the excluder peer. Also, there was a significant main effect of peer, $F(1, 33)$ = 4.73, $p$ = .037, $\eta p^2$ = .12, indicating overall differences in the responses for the includer and the excluder peer. The muscle X peer interaction was followed by posthoc analyses to investigate muscle differences between peers, averaged across emotions.

Paired samples t-tests revealed a significant difference in the muscle activation for the excluder peer, $t(33)$ = 2.84, $p$ = .008 between ZM ($M$ = 0.28, $SD$ = 0.62) and CS ($M$ = -0.15, $SD$ = 0.60), and no significant difference, $t(33)$ = -0.32, $p$ = .751, in muscle activation for the includer peer between ZM ($M$ = 0.20, $SD$ = 0.55) and CS ($M$ = 0.24, $SD$ = 0.46). These results suggest that while no significant facial reaction emerged for the includer peer, children reacted with a smile-like facial expression to the excluder peer, irrespective of the facial emotional display.

## Resistant and avoidant attachment modulation of facial mimicry

The modulation of happy and sad facial mimicry towards the excluder peer by resistant ($M$ = 15.48, S$D$ = 5.24, $Min$ = 4, $Max$ = 26) and avoidant ($M$ = 14.58, S$D$ = 4.19, $Min$ = 4, $Max$ = 26) attachment was analysed in two separate regression analyses. Resistant and avoidant attachment were not related ($r$ = .188, $p$ = .191). Prior to the analyses, the normality distribution of the attachment scores were examined with the Shapiro-Wilk test [53], which revealed a $W$ = .94, $p$ = .064 and a $W$ = .97, $p$ = .452 for resistant and avoidant attachment respectively, indicating that the data did not significantly deviate from a normal distribution. To compute the dependent variables, we calculated the difference scores between the ZM and CS mean activation for happy, whereas sad facial mimicry was calculated as the difference between CS and ZM mean activation. A positive difference score between the ZM and the CS represents a congruent response to happy, whereas a positive difference score between CS and ZM indicates a congruent response to sad expressions. The first analysis investigated the modulation of attachment tendencies on happy facial mimicry by regressing the scores of resistant, avoidant and the interaction between resistant and avoidant attachment patterns on the happy mimicry difference score. This analysis revealed no significant main effects of resistant, $\beta$ = -0.16, $t(3, 25)$ = -0.77, $p$ = .447, or avoidant patterns, $\beta$ = 0.07, $t(3, 25)$ = 0.32, $p$ = .749, and no significant interaction effects, $\beta$ = 0.03, $t(3, 25)$ = 0.14, $p$ = .888 on happy facial mimicry. Similarly, the results from the second analysis that investigated the modulation of attachment tendencies on sad facial mimicry by regressing the scores of resistant, avoidant and the interaction between resistant and avoidant attachment patterns on the sad mimicry difference score revealed no significant main effects of resistant, $\beta$ = 0.07, $t(3, 25)$ = 0.34, $p$ = .732, or avoidant patterns, $\beta$ = 0.15, $t(3, 25)$ = 0.72, $p$ = .479, and no significant interaction effects, $\beta$ = -0.08, $t(3, 25)$ = -0.37, $p$ = .713.

## Exploratory analyses of secure versus insecurely attached children's facial mimicry

After our planned regression analyses on the effect of insecure attachment tendencies (i.e. resistant and avoidant) on facial mimicry towards the excluder peer, we also explored facial mimicry differences between securely and insecurely attached children. It is possible that our null findings on the relationship between resistant and avoidant attachment tendencies and facial mimicry were caused by the low variability of avoidant and resistant attachment in our sample. Hence, we grouped participants into secure an insecure attachment, based on cut-off guidelines, irrespective of their specific tendency (resistant or avoidant), with the aim to contrast maximally different groups. To this end, the sample was split in two groups based on the

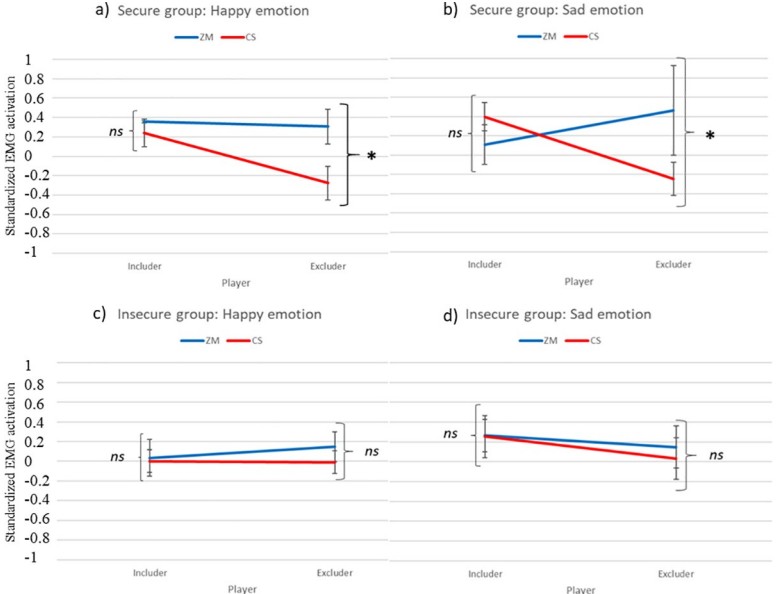

**Fig 4.** The figure illustrates the muscle activation in the ZM (blue) and the CS (red) muscle in the secure (a, b) and in the insecure group (c, d) for happy and sad emotion. *ns* = non-significant; * $p < .05$.

cut-off criteria of the AISI, with scores < 46 qualifying as secure attachment and scores > 46 as insecure attachment, which entail both resistant and avoidant (N = 20 secure; N = 14 insecure) tendencies. Noteworthy, the AISI instrument gives an indication of children's attachment tendencies, and not clinical categories. To test these differences, we conducted two repeated measures ANOVA analyses, one per emotion, with the factors muscle (ZM, CS) X peer (includer, excluder) X security (secure, insecure) and happy and sad facial mimicry as dependent variables. Results are illustrated in Fig 4.

For the sad emotion, the analyses yielded a significant muscle X peer X security interaction, $F(1, 32) = 4.24$, $p = .048$, $\eta p^2 = .12$. We followed up this result with two separate ANOVAs and found a significant muscle X peer interaction, $F(1, 19) = 11.94$, $p = .003$, $\eta p^2 = .39$, in the securely attached group, but not in the insecurely attached group, $F(1, 13) = 0.12$, $p = .736$. Paired sample t-tests showed that securely attached children displayed higher ZM ($M = 0.46$, $SD = 0.84$) compared to CS muscle activation ($M = -0.24$, $SD = 0.76$) for the sad expression, revealing a smiling response to the sad facial expression of the excluder peer, $t(19) = 3.18$, $p = .005$. Conversely, they reacted with a congruent facial expression to the sad emotion of the includer with higher CS ($M = 0.40$, $SD = 0.65$) than ZM muscle activation ($M = 0.11$, $SD = 0.92$), yet this difference did not reach statistical significance, $t(19) = 1.04$, $p = .311$. In the insecurely attached group, no significant differences emerged in their muscle facial responses to the sad emotion (all $ps > .29$).

Lastly, for the happy emotion, no significant muscle X peer X security interaction emerged, $F(1, 32) = 0.69$, $p = .413$.

## Discussion

The current study investigated whether young children's facial mimicry following social ostracism is modulated by attachment tendencies that underlies different affiliation motivations. For this purpose, we assessed facial EMG responses to happy and sad facial expressions of two peers with whom children had interacted via a computerized ball-tossing game and who had

either played fairly or excluded them. We expected that children would show stronger facial mimicry to a peer who had excluded them in the game as means to restore affiliation than to a peer who had played fairly. Furthermore, we hypothesized that attachment tendencies would modulate children's facial mimicry responses to the excluder peer. Specifically, we expected children with resistant tendencies to show enhanced facial mimicry and children with avoidant tendencies to show reduced facial mimicry of the excluder peer.

Our EMG results revealed no indication for differences in the facial reactions for happy and sad expressions towards the includer peer. Interestingly however, children reacted with a smile-like expression to the excluder's sad facial expression, indexed by increased zygomaticus activation. This response can be interpreted in different ways. It could be argued for instance that smiling to an unkind peer's sad expression represents a behavioural cue of interpersonal warmth that manifests through positive affect [63, 64] and thus may be an affiliative attempt. Relatedly, previous research revealed that smiles are related to approach behaviours [64–66]. Alternatively, children's smiles to the excluder's sad expression may interpreted as a retaliation response towards the unkind peer [67]. In accordance with this explanation, a series of experimental studies showed that social exclusion yields antisocial responses (e.g. in Cyberball or virtual chat room paradigms) [68–72]. More research is needed to investigate the cognitive and emotional processes behind children's smile to an excluder's sad emotion. For instance, previous evidence suggests that greater cognitive control and social understanding modulates behavioural mimicry responses towards an unkind confederate [6, 72].

We also hypothesized that attachment tendencies would modulate children's facial mimicry responses to the excluder peer. Our data showed that neither resistant nor avoidant attachment tendencies were significantly associated with facial mimicry responses to the excluder peer. This finding is in contrast to previous evidence indicating that 3-year-old children characterized by resistant attachment tendencies display enhanced facial mimicry to sad emotions [14]. These seemingly contradictory findings may be explained by a methodological difference between this and earlier research. In the current study, we investigated the modulation of attachment following a stressful situation of social ostracism, whereas the previous study by Vacaru and colleagues [14] only investigated children's intrinsic motivation for affiliation, stemming from early attachment relationships, irrespective of context. As such, it could be the case that children's motivation for affiliation in the context of social ostracism differs from and prevails over their intrinsic affiliation motivation, as determined by their early attachment relationships. It is also noteworthy that the children in our sample were not drawn from a clinical sample and thus disturbed attachment patterns were rare. Moreover, our sample consisted of children of parents who had indicated to be interested in participating in research together with their child. These parents might be especially interested in their children' development, which might also lead to them being more sensitive in the interaction with their children. Consequently, our sample was prevalently characterized by secure attachment (59%). Investigations across different populations, offering a broader spectrum of insecure attachment are needed to fully understand the relationship between mimicry and attachment. Besides, children played the Cyberball game on a computer while their parents were in the same room, and parents might have alleviated the effects of social exclusion by providing a safe environment through their mere presence. This could be especially true for securely attached children. For securely attached children, their parents' presence may constitute a safe haven [73] where children find refuge in a moment of distress (i.e. ostracism in the game). However, for insecurely attached children, the presence of their parents may not instil feeling safe, and hence their presence may not have alleviated the effects of the stressor. Accordingly, in our study, securely attached children might have reacted to the excluder peer with a smile possibly because they felt protected by their parent's presence in this ostracizing situation. The lack of a response in

the insecurely attached children, however, may indicate that they did not feel safe enough to react to the unpleasant situation. In the future, studies using live interactions between peers will allow for examining more ecological valid social dynamics and potentially test the effect of parent's proximity.

Nonetheless, the lack of significant relations between resistant and avoidant attachment tendencies with facial mimicry responses to the excluder peer could highlight a different strategy that insecurely attached children employ during stressful situations. Hence, we further investigated facial mimicry differences between securely and insecurely attached children. Interestingly, we found that securely attached children reacted with a smile to the excluder's sad facial expressions, whereas insecurely attached children did not show any facial responses, possibly withholding any reaction towards both peers. Within the conscience development framework, it has been suggested that attachment security is associated with conscience development in young children [74, 75] and that securely attached children may hold a greater sense of fairness and a greater number of socially competent solutions to social problems [76]. Therefore, they might be more likely to show punitive actions to others who violate expected norms (i.e. play fairly with peers) and thus display a punitive smile to the excluder peer's sad expression. Instead, the lack of a response from the insecurely attached children may suggest a freezing response as a defensive mechanism during a negative social interaction. In accordance with this, a longitudinal study has shown that early attachment insecurity is related to freezing-like behaviours (such as reduced heart rate and body sway) in adolescents when exposed to negative facial expressions [77]. This is also in line with the proposition that the inability to feel an emotion leads to lower expressiveness and reversely lower expressiveness induces the inability to feel [78, 79]. Such a response may be adaptive in insecurely attached children as means to prevent themselves from being overwhelmed by negative feelings and be able to assess the risks in the environment [80]. These findings contribute important preliminary evidence to the field of attachment by highlighting possible regulatory mechanisms employed by insecure children during negative social interactions. Consequently, this proposition might have implications across the lifespan and later development of psychopathology. Indeed, disengagement, like we saw in the insecure children, has been postulated as one central aspect for mistuned dyadic interactions and risk for mental health [81]. Further research is needed to unveil how psychological (i.e. attachment) and physiological (i.e. subtle mimicry, heart rate) factors interact in the modulation of social dynamics in normative and clinical samples.

Altogether, our study revealed that 5-year-olds displayed smiling facial reactions to an excluder but not an includer peer following social ostracism, irrespective of the emotional expression. Interestingly, only securely attached children showed these facial reactions. These findings partially substantiate the body of evidence that ostracism yields an increase in affiliative behaviours, here indexed by facial reactions to an excluder peer [27–29], and contribute new evidence on the modulation of affiliative facial reactions by individual differences. Moreover, the finding that securely attached children react with a smile to an excluder's sad facial expression raises questions regarding children's affiliative versus retaliation responses to social exclusion and their underlying socio-cognitive and emotional processes. This research further highlights the potential of novel techniques, such as facial electromyography to investigate subtle interpersonal dynamics following ostracism, in peer relationships.

## Supporting information

**S1 Table. Summary statistics of muscle activation: Descriptive and correlation analyses.** (DOCX)

## Acknowledgments

We sincerely thank Elisabeth Watzlawek, Rosa Pranger and Angela Khadar for help with participant recruitment and data collection, the members of our research group for fruitful discussions, and the parents and children for participating in the study.

## Author Contributions

**Conceptualization:** Stefania V. Vacaru, Johanna E. van Schaik, Erik de Water, Sabine Hunnius.

**Data curation:** Stefania V. Vacaru.

**Formal analysis:** Stefania V. Vacaru.

**Methodology:** Stefania V. Vacaru, Johanna E. van Schaik, Erik de Water, Sabine Hunnius.

**Project administration:** Stefania V. Vacaru.

**Resources:** Sabine Hunnius.

**Supervision:** Johanna E. van Schaik, Sabine Hunnius.

**Writing – original draft:** Stefania V. Vacaru.

**Writing – review & editing:** Stefania V. Vacaru.

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
