## [Decision Letter · Decision Letter 0]

13 May 2020

PONE-D-20-07094

Five-Year-Olds’ Facial Mimicry Following Social Ostracism is Modulated by Attachment Security

PLOS ONE

Dear Mrs Vacaru,

Thank you for submitting your manuscript to PLOS ONE. After careful consideration, we feel that it has merit but does not fully meet PLOS ONE’s publication criteria as it currently stands. Therefore, we invite you to submit a revised version of the manuscript that addresses the points raised during the review process.

Both reviewers considered the paper interesting and were generally positive but also had substantial theoretical and methodological comments that should be addressed in a revision.. . . 

We would appreciate receiving your revised manuscript by Jun 27 2020 11:59PM. To enhance the reproducibility of your results, we recommend that if applicable you deposit your laboratory protocols in protocols.io, where a protocol can be assigned its own identifier (DOI) such that it can be cited independently in the future. For instructions see: http://journals.plos.org/plosone/s/submission-guidelines#loc-laboratory-protocols

We look forward to receiving your revised manuscript.

Kind regards,

Peter A. Bos

Academic Editor

PLOS ONE

Journal Requirements:

2. You indicated that you had ethical approval for your study. In the ethics statement in the Methods and online submission information, please specify the name of the IRB that reviewed and approved your study; the review process at PLOS ONE is not anonymized.

3. We note that Figure 1 in your submission contain copyrighted images.

All PLOS content is published under the Creative Commons Attribution License (CC BY 4.0), which means that the manuscript, images, and Supporting Information files will be freely available online, and any third party is permitted to access, download, copy, distribute, and use these materials in any way, even commercially, with proper attribution. For more information, see our copyright guidelines: http://journals.plos.org/plosone/s/licenses-and-copyright.

We require you to either (a) present written permission from the copyright holder to publish this figure specifically under the CC BY 4.0 license, or (b) remove the figure from your submission:

b.    If you are unable to obtain permission from the original copyright holder to publish this figure under the CC BY 4.0 license or if the copyright holder’s requirements are incompatible with the CC BY 4.0 license, please either i) remove the figure or ii) supply a replacement figure that complies with the CC BY 4.0 license. Please check copyright information on all replacement figures and update the figure caption with source information. If applicable, please specify in the figure caption text when a figure is similar but not identical to the original image and is therefore for illustrative purposes only.

Reviewers' comments:

Reviewer's Responses to Questions

**Comments to the Author**

1. Is the manuscript technically sound, and do the data support the conclusions?

Reviewer #1: Partly

Reviewer #2: Partly

2. Has the statistical analysis been performed appropriately and rigorously? 

Reviewer #1: Yes

Reviewer #2: I Don't Know

3. Have the authors made all data underlying the findings in their manuscript fully available?

Reviewer #1: No

Reviewer #2: Yes

4. Is the manuscript presented in an intelligible fashion and written in standard English?

Reviewer #1: Yes

Reviewer #2: Yes

5. Review Comments to the Author

Reviewer #1: Thank you for the opportunity to review this well-written manuscript. This manuscript reports findings from an experimental facial electromyography study in [early] childhood. The study is very interesting, methodologically novel, and has the potential to make a significant contribution to our understanding of how attachment moderates children’s facial mimicry in response to social exclusion vs inclusion. However, I believe there are many major issues that need to be addressed by the authors.

Major Comments

1. Conceptual Issues:

Motivation and emotion literature show that facial mimicry is not only communicative, it is also reflective of inner affective and emotional states (Cacioppo, Petty, Losch, and Kim 1986; Hess et al., 2017; van Boxtel, 2010) that arise under specific social dynamics (Kraaijenvanger, Hofman, & Bos, 2017; Bos, Jap-Tjong, Spencer, & Hofman, 2016). This is an aspect of the current study that I see less highlighted at the present, but appears central. Findings show that individual differences in situation-relevant traits moderate facial reactions to such situations. Affiliative people smile more when viewing positive images of affiliation (Dufner, Hagemeyer, Arslan, Schönbrodt, & Denissen, 2015). Power-motivated people frown more when audiences are displeased by their impromptu speech (Fodor, Wick, & Hartsen, 2006), or when viewing videos with assertive persons in control (Fodor & Wick, 2006). Children with callous-unemotional traits smile when viewing film clips of anger, and exhibit lower increases in zygomaticus activity when viewing film clips of happiness (de Wied, van Boxtel, Matthys, & Meeus, 2011). In light of such findings, it is reasonable to think that the present study is mainly targeting children’s affect or empathy following ostracism rather than their compensatory friendliness, conformity, or submissiveness in the form of facial mimicry. This study actually resonates more with that literature, and so does the authors’ alternative interpretation of the findings (vindictive schadenfreude of securely attached children, affective indifference of insecurely attached ones). Following this line of reasoning, the current study can also build on the findings from Vacaru, van Schaik, Hunnius (2019) in order to structure the introduction and explain its findings. The authors could, if that helps, also consider that the inability to feel leads to lower expressiveness (and in a reciprocal manner, lower expressiveness also induces inability to feel; Niedenthal, Brauer, Halberstadt, & Innes-Ker, 2010; Oberman, Winkielman, & Ramachandran, 2007), and that corrugator activity can also signal concentration (Bos, Jap-Tjong, Spencer, & Hofman, 2016), so it is also possible that securely attached children withdrew their attention from pictures of excluding peers. Overall, I believe that the study will greatly benefit if it concretely acknowledges the central role of children’s own psychological states in responding to the images of two peers that they had very different social interactions with (a positive vs a negative one).

2. Methodological issues concerning attachment:

It is increasingly acknowledged that clinical entities and diagnoses, including pathological attachment patterns, fall into a spectrum. Not only does the current DSM and relevant attempts reflect that, but also the tool that the authors relied on. The AISI is a Likert scale, and so analyses would be more nuanced and precise if they were to primarily use the scale as a continuous, rather than only with cut-off scores (which, as the authors note, do not reflect diagnoses anyway and could be their secondary option). Note that this is something already resolved in the adult attachment literature (Fraley & Waller, 1998), which (like the AISI) also treats anxiety and avoidance as two individual difference dimensions. Furthermore, the authors could also examine separately the avoidance and the ambivalence/resistance subscales, because it is very intriguing from a conceptual perspective to know whether what drives the effects is attachment anxiety (i.e., ambivalence), attachment avoidance, or both. I am also uncertain whether an interaction term of ambivalence and avoidance when using cutoff scores gives the necessary power in such a small sample (the authors also mention the very few children above each of the insecurity cut-off scores). Finally, we are also missing the intercorrelation between the subscales, as well as their descriptives. To sum it up, the authors could treat their individual difference variable differently, and if not, mention why and how that might be a limitation.

3. Methodological issues concerning EMG recordings:

Whereas smiling usually involves higher zygomaticus and lower corrugator activity, there are cases when people smile but their corrugator does not drop. In fact, presenting correlations of the muscles in response to each type of image are, to me, important if the authors wish to continue with their current analytic approach. More importantly, however, these muscles are not always compared with each other in the analyses of many articles (e.g., many articles cited above). Based on these, I would carry out analyses separately for corrugator and zygomaticus, discuss what happens with each muscle separately, and then conceptually integrate everything. Therefore, I would suggest a different analytic approach, see if that makes everything more conceptually clear, and then examine whether the juxtaposition of muscle activities (which the authors do now) makes more sense or summarizes the findings better, because it relies on assumptions that are not always met. Related to that, I would not conclude that lower corrugator activity equals more smiling, when zygomaticus activity is constant (p.14).

4. Discussion:

I would like some suggestions for future research and a highlight of the study’s limitations, which are now completely absent from the discussion. This would allow the readers to see the study more objectively and to zoom out.

Minor Comments

1. The data correction and normalization process should be cited, because there are many different ways to preprocess data

2. Page 13, line 306-there is something missing there.

3. Given that the average age is almost 5 years, it is better to include age in years instead of months.

4. I would avoid using acronyms for the constructs of the study in the text (e.g. FM for facial mimicry), because it interferes with the flow of reading. I would still keep acronyms for methods (e.g., fMRI, EMG, etc.)

5. I would also include the term “ambivalent” for attachment in the introduction (p.4), especially since this is the first of the 2 terms (ambivalent/resistant), and perhaps briefly explain how that might fit with other, adult dimensional models of attachment (Bartholomew & Horowitz, 1991; Fraley & Waller, 1998), for terminological consistency with the broader attachment literature.

Reviewer #2: The current study investigates effects of 5-year olds’ attachment patterns on their facial mimicry/ subtle facial expressions toward peers that included or excluded them from a virtual game. The topic is interesting and the study seems basically sound. However, the paper has several shortcomings that need to be addressed, both in the presentation of the theoretical framework and methodological details, and in the interpretation and discussion of findings.

Introduction.

1) The hypothesized effects of attachment patterns on facial mimicry after social exclusion make sense. However, in presenting their argument the authors take a few shortcuts that are, in my opinion, not quite warranted. Most importantly, the authors equate attachment representations to affiliative motivation (even stating that they use attachment as a proxy for affiliation motives). Although it is undisputed that attachment representations are related to affiliative tendencies, attachment and affiliation motivation are not interchangeable constructs. The authors should not treat attachment as a proxy for affiliation, but rather argue why and how attachment representation would influence affiliative tendencies, and facial mimicry, in the expected direction.

2) Although I agree that it is reasonable to expect children to show stronger mimicry of the excluder’s than the includer’s expressions (p.6), I think the authors should (briefly) argue why, and not just state that it is in line with previous findings.

3) I wonder to what extent the authors expect effects of attachment to be specific to mimicry of the excluder (p.6) and not includer. Whether testing associations between attachment and mimicry of only the excluder is appropriate depends on the answer.

3) The authors should check the appropriateness of the references used throughout the Introduction. E.g. refs. 3-6 after ‘To achieve affiliation, individuals use subtle strategies from early on in development’ (p.3) are about the development of mimicry, ref. 29 (p.4) is to the entire handbook of attachment.

Methods

4) I think the Method section would be easier to read if topics were presented in a more conventional order (participants, procedure, instruments, analyses; also keep the mimicry paradigm and EMG recording and processing together).

5) The sample size seems adequate, but as justification the authors state ‘The sample size was justified in light of prior research employing facial EMG or social ostracism paradigms during childhood’ (p.7). I wonder what this means. Please elaborate. I also wonder whether a power analysis was conducted.

6) Please provide some more details on the cyberball game (p.7/8), e.g. the number of trials and task duration.

7) It is stated that ‘happy and sad facial expressions … were repeated eight times in pseudo-randomized order’ (bottom of page 8). Please specify what pseudo-randomized entails.

8) It would be good to cite some evidence for the reliability and validity of the Attachment Insecurity Screening Inventory (p. 9).

9) Minor detail: ‘a low cut-off of 10 Hz and a high cut-off of 1000 Hz’ (p.11, line 252) refers to a filter. Please state so.

10) EMG data processing (p.11/12). Some information is missing or unclear:

‘The remaining trials were filtered… (lines 260/261). Please explain ‘remaining trials’; were some removed already?

‘…band rejection filter of…’ (261). Please clarify whether the notch filter as implemented in BVA was used or something else.

A bandpass filter with cut-offs at 20 and 500Hz is mentioned. What was the slope?

‘… artifact rejection based on visual investigation…’ (264). Please provide some details regarding the criteria used.

‘EMG data was standardized … standard deviation of all bins’ (p. 12). I suppose the authors binned the data by computing the average EMG signal across each bin and a ‘value’ reflects an average in microvolts. Please specify (including the unit of measurement).

11) ‘… we ran two linear regression models with happy and sad FM as the dependent variables…’ (p.13, 295/296). I would prefer to read about the computation of happy and sad FM here rather than in the Results section. I also wonder why the authors chose to analyze the data this way only for this analysis. The use of dependent variables is inconsistent across analyses. This can be defended, but please explain why choices were made.

Results

12) Minor comment: The term ‘condition’ seems to be used interchangeably with ‘peer’ on p. 13. Please be consistent in wording.

13) I wonder whether the authors evaluated the distribution of attachment scores (normality, outliers) prior to performing regression analyses. Please elaborate.

14) After their hypotheses regarding attachment avoidance and resistance are not confirmed, the authors run exploratory analyses separating the sample into those securely and insecurely attached. Although performing exploratory analyses is not a problem as long as one is clear about it (which the authors are), I do not quite follow the justification provided. Please explain what is meant with ‘This analysis was justified by the sample…attachment dimensions’ (p.15).

15) P. 16: For the happy expression, the muscle*peer*security interaction is not significant. It is unsound to explore it further and interpret is as if it were significant. The corresponding results and conclusions should be removed from the paper (also from the Discussion and abstract).

Discussion

16) The Discussion focuses heavily on the interpretation of findings from the exploratory analyses. I wonder about the authors’ thoughts about their original hypotheses and why these were not confirmed. Please elaborate.

17) As the findings regarding attachment security were obtained in exploratory analyses, these might be an excellent starting point for future/follow-up research. I wonder what the authors think.

18) I think the authors should discuss some (of the consequences of the) limitations of their study.

6. PLOS authors have the option to publish the peer review history of their article (what does this mean?). If published, this will include your full peer review and any attached files.

Reviewer #1: No

Reviewer #2: No

---

## [Author Response · Author response to Decision Letter 0]

14 Jul 2020

PONE-D-20-07094: Five-Year-Olds’ Facial Mimicry Following Social Ostracism is Modulated by Attachment Security

Reviewer 1

Major Comments

1. Conceptual Issues:

Motivation and emotion literature show that facial mimicry is not only communicative, it is also reflective of inner affective and emotional states (Cacioppo, Petty, Losch, and Kim 1986; Hess et al., 2017; van Boxtel, 2010) that arise under specific social dynamics (Kraaijenvanger, Hofman, & Bos, 2017; Bos, Jap-Tjong, Spencer, & Hofman, 2016). This is an aspect of the current study that I see less highlighted at the present, but appears central. Findings show that individual differences in situation-relevant traits moderate facial reactions to such situations. Affiliative people smile more when viewing positive images of affiliation (Dufner, Hagemeyer, Arslan, Schönbrodt, & Denissen, 2015). Power-motivated people frown more when audiences are displeased by their impromptu speech (Fodor, Wick, & Hartsen, 2006), or when viewing videos with assertive persons in control (Fodor & Wick, 2006). Children with callous-unemotional traits smile when viewing film clips of anger, and exhibit lower increases in zygomaticus activity when viewing film clips of happiness (de Wied, van Boxtel, Matthys, & Meeus, 2011). In light of such findings, it is reasonable to think that the present study is mainly targeting children’s affect or empathy following ostracism rather than their compensatory friendliness, conformity, or submissiveness in the form of facial mimicry. This study actually resonates more with that literature, and so does the authors’ alternative interpretation of the findings (vindictive schadenfreude of securely attached children, affective indifference of insecurely attached ones). Following this line of reasoning, the current study can also build on the findings from Vacaru, van Schaik, Hunnius (2019) in order to structure the introduction and explain its findings. The authors could, if that helps, also consider that the inability to feel leads to lower expressiveness (and in a reciprocal manner, lower expressiveness also induces inability to feel; Niedenthal, Brauer, Halberstadt, & Innes-Ker, 2010; Oberman, Winkielman, & Ramachandran, 2007), and that corrugator activity can also signal concentration (Bos, Jap-Tjong, Spencer, & Hofman, 2016), so it is also possible that securely attached children withdrew their attention from pictures of excluding peers. Overall, I believe that the study will greatly benefit if it concretely acknowledges the central role of children’s own psychological states in responding to the images of two peers that they had very different social interactions with (a positive vs a negative one).

Response: We thank the reviewer for their suggestions, which we integrated in our introduction (see page 3, lines 53-59):

“Facial mimicry of emotional expressions is not merely the result of activation matching, but is also influenced by one’s affect. Contrary to postures, facial emotions are inherently meaningful and their mimicry engage affective and motivational processes [15–17]. Accordingly, inner states and intrinsic characteristics (e.g. attachment orientation [14]; callous-unemotional traits [18], in concert with affiliation motives [19]; power motivation [20] modulate the extent to which individuals manifest facial mimicry in social contexts [21,22].”

In addition, we also integrated the reviewer’s suggestions in our discussion, by editing the text as follows (see page 21, lines 512-514): 

“This is also in line with the proposition that the inability to feel an emotion leads to lower expressiveness and reversely lower expressiveness induces the inability to feel [79,80]. Such a response may be adaptive in insecurely attached children as means to prevent themselves from being overwhelmed by negative feelings and be able to assess the risks in the environment [69].”

2. Methodological issues concerning attachment:

It is increasingly acknowledged that clinical entities and diagnoses, including pathological attachment patterns, fall into a spectrum. Not only does the current DSM and relevant attempts reflect that, but also the tool that the authors relied on. The AISI is a Likert scale, and so analyses would be more nuanced and precise if they were to primarily use the scale as a continuous, rather than only with cut-off scores (which, as the authors note, do not reflect diagnoses anyway and could be their secondary option). Note that this is something already resolved in the adult attachment literature (Fraley & Waller, 1998), which (like the AISI) also treats anxiety and avoidance as two individual difference dimensions. Furthermore, the authors could also examine separately the avoidance and the ambivalence/resistance subscales, because it is very intriguing from a conceptual perspective to know whether what drives the effects is attachment anxiety (i.e., ambivalence), attachment avoidance, or both. I am also uncertain whether an interaction term of ambivalence and avoidance when using cut-off scores gives the necessary power in such a small sample (the authors also mention the very few children above each of the insecurity cut-off scores). Finally, we are also missing the intercorrelation between the subscales, as well as their descriptives. To sum it up, the authors could treat their individual difference variable differently, and if not, mention why and how that might be a limitation.

Response: There seem to be a misunderstanding regarding the attachment variables used for the main analyses. In the main analyses of the current study (i.e. the regression models), we did use the continuous scores rather than the cut-off scores, to examine the main effects and the interaction between resistant/ambivalent and avoidant attachment on facial mimicry. In the regression models, we assessed the contribution of resistant/ambivalent and avoidant attachment to facial mimicry separately, by looking at the variance explained by each main effect in the model. Accordingly, we examined the contribution of the avoidance and resistance subscales to facial mimicry following the procedure described by Vacaru et al. (2019). Only in the exploratory analyses we performed the RM ANOVA with security as a between-subjects factor with two levels. We added the descriptives of attachment and avoidant attachment as well as their correlation (page 15, lines 374-377): “The modulation of happy and sad facial mimicry towards the excluder peer by resistant (M = 15.48, SD = 5.24, Min = 4, Max = 26) and avoidant (M = 14.58, SD = 4.19, Min = 4, Max = 26) attachment was analyzed in two separate regression analyses. Resistant and avoidant attachment were not related (r = .188, p = .191).”

3. Methodological issues concerning EMG recordings:

Whereas smiling usually involves higher zygomaticus and lower corrugator activity, there are cases when people smile but their corrugator does not drop. In fact, presenting correlations of the muscles in response to each type of image are, to me, important if the authors wish to continue with their current analytic approach. More importantly, however, these muscles are not always compared with each other in the analyses of many articles (e.g., many articles cited above). Based on these, I would carry out analyses separately for corrugator and zygomaticus, discuss what happens with each muscle separately, and then conceptually integrate everything. Therefore, I would suggest a different analytic approach, see if that makes everything more conceptually clear, and then examine whether the juxtaposition of muscle activities (which the authors do now) makes more sense or summarizes the findings better, because it relies on assumptions that are not always met. Related to that, I would not conclude that lower corrugator activity equals more smiling, when zygomaticus activity is constant (p.14).

Response: We agree that activation or deactivation of a muscle compared to baseline does not necessarily indicate a smile or a sad expression. This is why we look at the facial configuration by means of the composite score resulting from the subtraction of the activation in the non-corresponding muscle from the corresponding muscle. A facial expression is the result of a multitude of muscles, but as we cannot measure activation of all muscles involved, we compare activation in the most important muscles with each other. This approach also reduces the number of factors in our subsequent analyses. Although in the literature there are different approaches to measure facial mimicry, we adopted the composite score approach that is commonly used in studies examining facial mimicry in relation to another dependent variable (i.e. ADHD, attachment; e.g. Deschamps et al., 2012; Deschamps et al., 2014; de Klerk et al., 2018; Vacaru et al., 2019). For completeness, we followed the suggestion of the reviewer to examine the intercorrelations among the muscles. The results of this analysis are now given in the supplementary materials (see S1). 

4. Discussion:

I would like some suggestions for future research and a highlight of the study’s limitations, which are now completely absent from the discussion. This would allow the readers to see the study more objectively and to zoom out.

Response: Thank you for this observation. We now addressed this issue by adding the following sentences to the Discussion section (see page 20, lines 483-495): “It is also noteworthy that the children in our sample were not drawn from a clinical sample and thus disturbed attachment patterns were rare. Moreover, our sample consisted of children of parents who had indicated to be interested in participating in research together with their child. These parents might be especially interested in their babies’ development, which might also lead to them being more sensitive in the interaction with their babies. Consequently, our sample was prevalently characterized by secure attachment (59%). Investigations across different populations, offering a broader spectrum of insecure attachment are needed to fully understand the relationship between mimicry and attachment. Besides, children played the Cyberball game on a computer while their parents were in the same room, and parents might have alleviated the effects of social exclusion by providing a safe environment through their mere presence. In the future, studies using live interactions between peers will allow for examining more ecological valid social dynamics.”

Minor Comments

1. The data correction and normalization process should be cited, because there are many different ways to preprocess data

Response: We clarified the guidelines used for EMG signal normalization as well as the slope (a 12 dB/octave slope) describing the filters used. The following sentence was added to the EMG data processing section (see page 12, lines 297-300): “EMG data were pre-processed and normalized with Brain Vision Analyzer 2.1 (BVA, 47), following the recommendations of De Luca and colleagues (53).”

2. Page 13, line 306-there is something missing there.

Response: We completed the sentence (see page 16, line 393): This analysis revealed no significant main effects of resistant, β = -0.16, t(3, 25) = -0.77, p = .447, or avoidant patterns, β = 0.07, t(3, 25) = 0.32, p = .749, and no significant interaction effects, β = 0.03, t(3, 25) = 0.14, p = .888, on happy facial mimicry.”

3. Given that the average age is almost 5 years, it is better to include age in years instead of months.

Response: We recalculated the age in years (page 7, lines 161-163): “Thus, a total of 34 children (21 girls; Mage = 4.7 years, SDage = .34 years, range = 3.81 – 5.27 years) were included in the final analyses.”

4. I would avoid using acronyms for the constructs of the study in the text (e.g. FM for facial mimicry), because it interferes with the flow of reading. I would still keep acronyms for methods (e.g., fMRI, EMG, etc.)

Response: We changed FM to facial mimicry throughout the manuscript.

5. I would also include the term “ambivalent” for attachment in the introduction (p.4), especially since this is the first of the 2 terms (ambivalent/resistant), and perhaps briefly explain how that might fit with other, adult dimensional models of attachment (Bartholomew & Horowitz, 1991; Fraley & Waller, 1998), for terminological consistency with the broader attachment literature.

Response: We clarified the terminology, also within the adult attachment framework, by editing the sentence as follows (page 4, lines 79-81): “Particularly, childhood insecure attachment can be distinguished into resistant/ambivalent (also called preoccupied in adulthood) and avoidant (also called dismissing in adulthood) [41]”.

Reviewer2

Introduction.

1) The hypothesized effects of attachment patterns on facial mimicry after social exclusion make sense. However, in presenting their argument the authors take a few shortcuts that are, in my opinion, not quite warranted. Most importantly, the authors equate attachment representations to affiliative motivation (even stating that they use attachment as a proxy for affiliation motives). Although it is undisputed that attachment representations are related to affiliative tendencies, attachment and affiliation motivation are not interchangeable constructs. The authors should not treat attachment as a proxy for affiliation, but rather argue why and how attachment representation would influence affiliative tendencies, and facial mimicry, in the expected direction.

Response: We thank the reviewer for this observation and now refrain from defining attachment as a proxy for affiliation. Instead, we now emphasize that the two attachment tendencies lead to specific strategies to achieve closeness and affiliation with others, as suggested also in the studies by Schwartz et al. (2007) and Vacaru et al. (2019). We therefore adapted the text (see page 7, lines 153-155): “Hence, we assessed resistant and avoidant attachment tendencies as they lead to distinct strategies to attain affiliation.”

2) Although I agree that it is reasonable to expect children to show stronger mimicry of the excluder’s than the includer’s expressions (p.6), I think the authors should (briefly) argue why, and not just state that it is in line with previous findings.

Response: We now elaborate on our hypothesis and clarify why stronger mimicry was expected for the excluder compared to the includer peer (see page 6, lines 131-134): “In other words, being ostracized will trigger their motivation to restore positive feelings and hence to affiliate with the excluder peer by means of facial mimicry, while this will not be the case for the includer peer, with whom the interaction was already of an affiliative inclusive nature.”

3) I wonder to what extent the authors expect effects of attachment to be specific to mimicry of the excluder (p.6) and not includer. Whether testing associations between attachment and mimicry of only the excluder is appropriate depends on the answer.

Response: We thank the reviewer for this comment. We now clarified the specificity of our hypothesis regarding the effect of attachment on facial mimicry towards the excluder peer. As such, we addressed this point as following (pages 6-7, lines 139-150): “According to the attachment theory, attachment behaviours become manifest under conditions of distress [50], which in this experiment is induced through the experience of ostracism. Attachment threat paradigms have been commonly used in attachment research to activate the attachment behavioural system and study the effects of different attachment tendencies on social dynamics, including mimicry [43,51,52]. Similarly, we expected that the effect of ostracism would be particularly salient in children characterized by insecure attachment, and resistant and avoidant tendencies would lead to different behavioural strategies. Moreover, as the attachment system is activated following a distressing experience, we expected that insecurely attached children would exhibit distinct dis/affiliative behaviours, aimed at restoring positive feelings in relation to the source of distress. Consequently, in line with previous evidence (14,36), we hypothesized that children characterized by resistant attachment would show increased facial mimicry to the excluder, as opposed to children characterized by avoidant attachment who were expected to show decreased facial mimicry, as means to increase or decrease chances for affiliation, respectively [23]. Hence, we assessed resistant and avoidant attachment tendencies as they lead to distinct strategies to attain affiliation.”

4) The authors should check the appropriateness of the references used throughout the Introduction. E.g. refs. 3-6 after ‘To achieve affiliation, individuals use subtle strategies from early on in development’ (p.3) are about the development of mimicry, ref. 29 (p.4) is to the entire handbook of attachment.

Response: Thank you for noticing! We have now addressed this issue by correcting and checking our references.

Methods

5) I think the Method section would be easier to read if topics were presented in a more conventional order (participants, procedure, instruments, analyses; also keep the mimicry paradigm and EMG recording and processing together).

Response: We re-arranged the methods section as suggested.

6) The sample size seems adequate, but as justification the authors state ‘The sample size was justified in light of prior research employing facial EMG or social ostracism paradigms during childhood’ (p.7). I wonder what this means. Please elaborate. I also wonder whether a power analysis was conducted.

Response: The sample size was determined with a power analysis assuming a medium effect size (based on previous research on the effect of attachment on facial mimicry; Vacaru et al., 2019), which yielded a sample size of N=36 with .95 power and α of 0.05. We added the following sentence to the manuscript (see page 7, lines 164-167): “We performed a power analysis that yielded a sample size of 36 participants based on a previously reported medium-sized effect of attachment on facial mimicry in children (β =.39; 14) with a power of .95 and α of 0.05 (GPower software; 52).”

7) Please provide some more details on the cyberball game (p.7/8), e.g. the number of trials and task duration.

Response: We have added more detail about the Cyberball paradigm to the Methods section, where we now report the average ball tosses and the average duration of the game (page 10, lines 238-242): “The number of ball tosses of the includer player was dependent on the tosses of the participant and to whom they tossed the ball first, as the includer had to toss the ball an equal number of times to each player. Overall, children tossed the ball 7.39 times (SD = 4.36) to the includer and 8.58 times (SD = 2.17) to the excluder, t(32) = 1.24, p = .222.”. We also added (pages 9-10, lines 248-250): “At the end, children played another round in which both players tossed the ball equally frequently to the participant to restore positive feelings. Altogether, the game lasted approximately 5 minutes.” 

8) It is stated that ‘happy and sad facial expressions … were repeated eight times in pseudo-randomized order’ (bottom of page 8). Please specify what pseudo-randomized entails.

Response: We clarified as follows (see page 11, lines 276-277): “In the facial mimicry task, happy and sad facial expressions of both models were shown eight times in a pseudo-randomized order with the constraints that the same peer displaying the same emotion was never repeated right after each other.”

9) It would be good to cite some evidence for the reliability and validity of the Attachment Insecurity Screening Inventory (p. 9).

Response: We now elaborated in the Method section on the validity of the instrument by adding the following (see page 9, lines 214-217): “Its reliability and validity have been examined across typical and clinical populations, indicating sound psychometric properties in terms of convergent, concurrent, and predictive validity as well as discriminating power between secure and insecure preschoolers (43).”

10) Minor detail: ‘a low cut-off of 10 Hz and a high cut-off of 1000 Hz’ (p.11, line 252) refers to a filter. Please state so.

Response: We clarified as follows (page 12, line 292): “A sampling rate of 2500 Hz was used, and a low cut-off and high cut-off filters of 10 Hz and 1000 Hz, respectively, were set for the data acquisition.”

11) EMG data processing (p.11/12). Some information is missing or unclear:

‘The remaining trials were filtered… (lines 260/261). Please explain ‘remaining trials’; were some removed already?

Response: We corrected the sentence by deleting the word “remaining”. No trials removed before the data preprocessing in Brain Vision Analyzer.

12) ‘…band rejection filter of…’ (261). Please clarify whether the notch filter as implemented in BVA was used or something else.

Response: We clarified the sentence as follows (page 12, lines 299-305): “EMG data were pre-processed with Brain Vision Analyzer 2.1. The trials were filtered using a notch filter with band rejection of 50 Hz, 0.2 bandwidth, order 4, as implemented in BVA.”

13) A bandpass filter with cut-offs at 20 and 500Hz is mentioned. What was the slope?

‘… artifact rejection based on visual investigation…’ (264). Please provide some details regarding the criteria used.

Response: We clarified this part by adding the information on the slope, according to recommendations by De Luca and colleagues (2010). Moreover, we elaborated on the criteria used for the artifact rejection (see page 13, lines 308-311): “(…) a 12 dB/octave slope was applied. (…) The signals of both muscles were screened between 500 ms prior and the 2000 ms after stimulus onset, hence for segments with a total duration of 2500 ms. The segments were inspected for extreme amplitude values outside a 100 mV range. If any peaks during a segment indicated such extreme values, the trial was rejected.”

14) ‘EMG data was standardized … standard deviation of all bins’ (p. 12). I suppose the authors binned the data by computing the average EMG signal across each bin and a ‘value’ reflects an average in microvolts. Please specify (including the unit of measurement).

Response: We clarified the unit of measure for the EMG mean activation, as follows (page 13, line 320): “To this end, data was divided in bins of 100 ms and each bin was standardized by subtracting from each value the mean activation in microvolts of all the bins and dividing it by the standard deviation of all bins.”

15) ‘… we ran two linear regression models with happy and sad FM as the dependent variables…’ (p.13, 295/296). I would prefer to read about the computation of happy and sad FM here rather than in the Results section. I also wonder why the authors chose to analyze the data this way only for this analysis. The use of dependent variables is inconsistent across analyses. This can be defended, but please explain why choices were made.

Response: Thank you for the suggestion, we have now moved this part to the section right prior to the description of the regression models in the Results.

For the regression analysis we had to compute a composite score of happy and sad mimicry as our DV in order to be able to investigate the relation between attachment and facial mimicry of happy and sad, also as seen in the study by Vacaru and colleagues (2019).

Results

16) Minor comment: The term ‘condition’ seems to be used interchangeably with ‘peer’ on p. 13. Please be consistent in wording.

Response: We adapted the text and replaced the word “condition” with “peer“ throughout.

17) I wonder whether the authors evaluated the distribution of attachment scores (normality, outliers) prior to performing regression analyses. Please elaborate.

Response: We have now reported the results from the normality distribution tests Shapiro-Wilk, which suggest that the scores of resistant and avoidant attachment are normally distributed. The following sentence was added in the text (see page 15, lines 379-385): “Prior to the analyses, the normality distribution of the attachment scores were examined with the Shapiro-Wilk test (54), which revealed a W = .94, p = .064 and a W = .97, p = .452 for resistant and avoidant attachment respectively, indicating that the data did not significantly deviate from a normal distribution.”

18) After their hypotheses regarding attachment avoidance and resistance are not confirmed, the authors run exploratory analyses separating the sample into those securely and insecurely attached. Although performing exploratory analyses is not a problem as long as one is clear about it (which the authors are), I do not quite follow the justification provided. Please explain what is meant with ‘This analysis was justified by the sample…attachment dimensions’ (p.15).

Response: Given the low variability in the attachment scores across the avoidance and resistance dimensions, we decided to pursue a different way of interpreting participants’ scores. Accordingly, we made use of the recommended cut-off scores that distinguish between secure and insecure attachment. We now clarified this in the manuscript by adding the following (page 17, lines 408-412): “It is possible that our null findings on the relationship between resistant and avoidant attachment tendencies and facial mimicry were caused by the low variability of avoidant and resistant attachment in our sample. Hence, we grouped participants into secure an insecure attachment, based on cut-off guidelines, irrespective of their specific tendency (resistant or avoidant), with the aim to contrast maximally different groups.”

19) P. 16: For the happy expression, the muscle*peer*security interaction is not significant. It is unsound to explore it further and interpret is as if it were significant. The corresponding results and conclusions should be removed from the paper (also from the Discussion and abstract).

Response: We have now removed the follow-up analyses for the muscle*peer*security interaction and their interpretation from the paper.

Discussion

20) The Discussion focuses heavily on the interpretation of findings from the exploratory analyses. I wonder about the authors’ thoughts about their original hypotheses and why these were not confirmed. Please elaborate.

Response: This suggestion has also been made earlier by Reviewer 1, point 4 (Discussion). We have addressed these suggestions by adding the following sentences to the manuscript (see page 20, lines 483-495): “It is also noteworthy that the children in our sample were not drawn from a clinical sample and thus disturbed attachment patterns were rare. Moreover, our sample consisted of children of parents who had indicated to be interested in participating in research together with their child. These parents might be especially interested in their babies’ development, which might also lead to them being more sensitive in the interaction with their babies. Consequently, our sample was prevalently characterized by secure attachment (59%). Investigations across different populations, offering a broader spectrum of insecure attachment are needed to fully understand the relationship between mimicry and attachment. Besides, children played the Cyberball game on a computer while their parents were in the same room, and parents might have alleviated the effects of social exclusion by providing a safe environment through their mere presence. In the future, studies using live interactions between peers will allow for examining more ecological valid social dynamics.” 

21) As the findings regarding attachment security were obtained in exploratory analyses, these might be an excellent starting point for future/follow-up research. I wonder what the authors think.

Response: We thank the reviewer for this suggestion. We added the following sentences to the Discussion section of the manuscript (see page 21, lines 518-527): “These findings contribute important preliminary evidence to the field of attachment by highlighting possible regulatory mechanisms employed by insecure children during negative social interactions. Consequently, this proposition might have implications across the lifespan and later development of psychopathology. Indeed, disengagement, like we saw in the insecure children, has been postulated as one central aspect for mistuned dyadic interactions and risk for mental health [81]. Further research is needed to unveil how psychological (i.e. attachment) and physiological (i.e. subtle mimicry, heart rate) factors interact in the modulation of social dynamics in normative and clinical samples.”

22) I think the authors should discuss some (of the consequences of the) limitations of their study.

Response: The paragraph that we added in response to the reviewers’ request for a more thorough discussion of our study’s limitations (see Reviewer 1’s point 4) and of the unexpected results (see Reviewer 2’s point 20) also contains our thoughts on the implications that follow from the unexpected results and limitations of our study (see page 20, lines 483-495):

“It is also noteworthy that the children in our sample were not drawn from a clinical sample and thus disturbed attachment patterns were rare. Moreover, our sample consisted of children of parents who had indicated to be interested in participating in research together with their child. These parents might be especially interested in their babies’ development, which might also lead to them being more sensitive in the interaction with their babies. Consequently, our sample was prevalently characterized by secure attachment (59%). Investigations across different populations, offering a broader spectrum of insecure attachment are needed to fully understand the relationship between mimicry and attachment. Besides, children played the Cyberball game on a computer while their parents were in the same room, and parents might have alleviated the effects of social exclusion by providing a safe environment through their mere presence. In the future, studies using live interactions between peers will allow for examining more ecological valid social dynamics.”

---

## [Editor Report · Decision Letter 1]

22 Sep 2020

PONE-D-20-07094R1

Five-Year-Olds’ Facial Mimicry Following Social Ostracism is Modulated by Attachment Security

PLOS ONE

Dear Dr. Vacaru,

Thank you for submitting your manuscript to PLOS ONE. After careful consideration, we feel that it has merit but does not fully meet PLOS ONE’s publication criteria as it currently stands. Therefore, we invite you to submit a revised version of the manuscript that addresses the points raised during the review process.

Although most of the comments were properly addressed, there are some remaining issues brought up by reviewer #2 that I need more attention. These are listed below this letter.

My apologies for the delay in this revision round, which was caused by the impact of Corona on the availability of both myself and the reviewers. 

We look forward to receiving your revised manuscript.

Kind regards,

Peter A. Bos

Academic Editor

PLOS ONE

Additional Editor Comments (if provided):

-  6: I am not sure what the issue is, as the authors do not mention the test that power was computed for, but even for a one-sided test of a .40 correlation at alpha = .05 a sample of n=59 is needed to achieve 95% power (power at n=36 is approximately .82), and the analyses that are actually conducted are more complex.

- 15: I think the authors should do more to address this point. The use of varying dependent variables can be defended, but please explain why choices were made and add those to the paper.

- o.a. 22: The authors have added a discussion of limitations to the Discussion section. However, there is one limitation that I think requires some more attention. The authors state that parents were present while children played the cyberball game and that parents’ presence may have affected children’s coping with social exclusion. The authors would do well to consider potential differential effects of the presence of parents on securely and insecurely attached children. Also, the parents’ presence should be mentioned in the Method section (e.g. under procedures).

As a minor comment: The authors refer to parents as parents of babies. They might change ‘babies’ to ‘children’.

---

## [Author Response · Author response to Decision Letter 1]

30 Sep 2020

Reviewer2

Methods

6) I am not sure what the issue is, as the authors do not mention the test that power was computed for, but even for a one-sided test of a .40 correlation at alpha = .05 a sample of n=59 is needed to achieve 95% power (power at n=36 is approximately .82), and the analyses that are actually conducted are more complex.

Response: The sample size was determined based on prior work using a similar approach in which a medium effect size (R2 = .39) of attachment on facial mimicry was found (Vacaru et al., 2019). Additionally, a power analysis was performed using a linear multiple regression from the t-test family, two-tailed, 3 predictors (i.e. resistant attachment, avoidant attachment, and their interaction) assuming a medium effect size of .39, with .95 power and α of 0.05. This calculation showed a required sample of 36 participants. We added this explanation to the manuscript (see page 7, lines 163-168).

15) The use of dependent variables is inconsistent across analyses. This can be defended, but please explain why choices were made and add those to the paper.

Response: In the first analysis, we investigated whether children display mimicry responses at all, and hence we employed a repeated measure ANOVA with emotion (happy, sad), muscle (CS, ZM) and peer (includer, excluder) as independent variables and mean standardized muscle activations as the dependent variables. This analysis provides information as to whether muscle activation differences exist as a function of the observed emotional expression, displayed by one or the other peer. 

 In the second analysis, however, our research question was different. We investigated the effect of attachment on the magnitude of the emotional facial response to each peer. To this end, we collapsed the two different muscles into one difference score as an index of a facial configuration: i.e. happy or sad. This choice is also in line with previous work, which first describes an ANOVA approach to investigate muscle activation as a function of the observed emotion, and then creates composite scores for each emotion based on the differences of the selected muscles (Deschamps et al., 2012, 2014; Vacaru et al., 2019). To clarify this point in the manuscript, we added this explanation to the Statistical Analyses section (see page 14, lines 329-338).

22) The authors have added a discussion of limitations to the Discussion section. However, there is one limitation that I think requires some more attention. The authors state that parents were present while children played the cyberball game and that parents’ presence may have affected children’s coping with social exclusion. The authors would do well to consider potential differential effects of the presence of parents on securely and insecurely attached children. Also, the parents’ presence should be mentioned in the Method section (e.g. under procedures).

Response: We thank the reviewer for this suggestion, which we have now given more attention in the discussion. We agree that the parent’s presence in the room may have had a different effect on securely and insecurely attached children. That is, for securely attached children their parents’ presence may constitute a safe haven (Powell et al., 2009) where children find refuge in a moment of distress (i.e. ostracism in the game). However, for insecurely attached children, the presence of their parents in the room may not instill more safety feelings in the children compared to if they were alone, and hence this may not have alleviated the effects of the stressor. Accordingly, in our study, securely attached children reacted to the excluder peer, possibly also because they felt protected by their parents’ presence in this ostracizing situation. The lack of a response in the insecurely attached children, in contrast, may indicate that they did not feel safe enough to react to the unpleasant situation. We added this explanation in the manuscript (see page 20, lines 476-474).

As a minor comment: The authors refer to parents as parents of babies. They might change ‘babies’ to ‘children’.

 Response: Thank you for noticing this. We have now replaced ‘babies’ with ‘children’ throughout the manuscript.

---

## [Editor Report · Decision Letter 2]

1 Oct 2020

Five-Year-Olds’ Facial Mimicry Following Social Ostracism is Modulated by Attachment Security

PONE-D-20-07094R2

Dear Dr. Vacaru,

We’re pleased to inform you that your manuscript has been judged scientifically suitable for publication and will be formally accepted for publication once it meets all outstanding technical requirements.

Kind regards,

Peter A. Bos

Academic Editor

PLOS ONE
---

## [Editor Report · Acceptance letter]

11 Dec 2020

PONE-D-20-07094R2 

Five-Year-Olds’ Facial Mimicry Following Social Ostracism is Modulated by Attachment Security 

Dear Dr. Vacaru:

I'm pleased to inform you that your manuscript has been deemed suitable for publication in PLOS ONE. Congratulations! Your manuscript is now with our production department. 

Kind regards, 

on behalf of

Dr. Peter A. Bos 

Academic Editor

PLOS ONE